# Organized Breast and Cervical Cancer Screening: Attendance and Determinants in Rural China

**DOI:** 10.3390/ijerph19148237

**Published:** 2022-07-06

**Authors:** Huinan Han, Xiaoyu Wang, Yimei Zhu, Yuan Liang

**Affiliations:** 1Department of Social Medicine and Health Management, School of Public Health, Tongji Medical College, Huazhong University of Science and Technology, 13 Hangkong Road, Wuhan 430030, China; hanhuinan1103@163.com (H.H.); wangxiaoyu361@163.com (X.W.); 2School of Media, Communication and Sociology, University of Leicester, Leicester LE1 7RH, UK; yz411@leicester.ac.uk

**Keywords:** organized cancer screening, organizer-level factors, public sector involvement, China

## Abstract

To evaluate the attendance and determinants of organized cervical and breast cancer (two-cancer) screening, especially higher-level factors, we conducted a cross-sectional survey in central China from June 2018 to November 2019 among 1949 women (age ≥ 35 years). We examined organizer-level factors, provider-level factors, receiver-lever factors and attendance and participation willingness of screening. The results indicate that the attendance and participation willingness of organized two-cancer screening was 61.19% and 77.15%, respectively. After adjustment for potential confounders, women who received screening notification were more likely to have greater participation willingness and higher attendance than those who received no notification (adjusted odds ratio [aOR] = 1.59, 95% confidence interval [CI]: 1.27–1.99; aOR = 98.03, 95% CI: 51.44–186.82, respectively). Compared with being notified about screening by GPs, being notified by community women’s leaders and other community leaders were more likely to lead to greater willingness to participate again (aOR = 2.86, 95% CI: 1.13–7.24; aOR = 3.27, 95% CI: 1.26–8.48, respectively) and recommending screening to others (aOR = 2.18, 95% CI: 1.02–4.65; aOR = 4.14, 95% CI: 1.84–9.30, respectively). The results suggest that notification of women about screening by community leaders is an important organizer-level factor. As a part of public health services, the design and implementation of optimal cancer screening strategies may require public-sector involvement at the organizer level instead of a one-man show by the health sector.

## 1. Introduction

Cancer incidence has gradually increased with the development of societies and economies. Cervical cancer and breast cancer (referred to in China as “two-cancer”) are the two most prevalent cancers among women, and the upward trend of incidence of these cancers is much greater among younger than older women [1,2]. Enhancing effective screening is indispensable for the reduction in two-cancer incidence and mortality [3,4]. Compared with opportunistic screening, organized screening is considered to have several potential advantages, including eliminating socioeconomic and ethnic disparities, increasing participation rates and population coverage, as well as increasing follow-up and quality control [5,6,7]. Thus, implementing organized screening programs is recommended in many countries and regions. However, the effect is not optimal. According to the 2015 national health interview survey in the United States, the screening rate for breast cancer within the preceding 2 years was 64.3% (age ≥ 40 years), and that of cervical cancer was 81.6% (age 21–64). Additionally, many women either have never been screened or are not screened regularly [8]. Data from Europe show great variability in two-cancer screening rates among countries, with the lowest less than 10% and the highest more than 80%. Most countries have not reached the 70% participation threshold, and no countries have achieved the 85% outlined in the European guidelines [9,10]. Data from 55 low- and middle-income countries during 2005–2018 also show wide variation in the self-reported lifetime prevalence of cervical cancer screening among countries, with a 44% median prevalence. Taken together, these data support the need to improve organized screening [11].

Since the new round of medical reform in 2009, China has included two-cancer screenings in the national public health services program. A nationally organized two-cancer screening program targeting women aged 35 years and above was launched, with implementation mechanisms that include government-led, multi-department cooperation; regional medical resource integration; and participation by the whole of society [12,13]. Specifically, screening costs are paid by central and local governments; screening notification for each household is mainly provided by GPs in village clinics (the vast majority of them are private) or by village committees. The latter is the grassroots autonomous organization with semi-official characteristics and with leaders elected by rural villagers. The members of a village committee are called “village cadres” and usually include the party branch secretary, director, and several vice-directors who are in charge of finance, public security, and women’s affairs; the latter is often called the women’s director. In the Chinese language, the term “village” means rural community, and in fact, village cadres are community leaders. Prior data show that the lowest cervical cancer screening rate in China is 28%, and the highest rate is 63% [14]. According to the Healthy China Initiative (2019–2030), the overall two-cancer screening rate was 52.6% in 2019 and will reach 80% or more and 90% or more by 2022 and 2030, respectively [15,16].

Conceptual models suggest that variation in attendance at two-cancer screening occurs at multiple levels, especially involving the receiver (individual women), provider, and organizer of screening; most of the literature has focused on receiver-related factors [17,18,19,20]. Younger, being unmarried, and having no children predict lower screening attendance, and previous experience with cervical abnormalities substantially predicts higher screening attendance. However, the association of education level, income, type of employment, and knowledge of screening with attendance are inconsistent among existing studies [20,21,22,23,24]. Provider-related factors, such as lack of GPs, educational intervention, and recommendations, as well as the distance to the screening site, have been associated with low screening attendance [20,21]. However, few studies have empirically measured providers’ performance with respect to screening attendance. Given the multiple steps and interfaces involved in organized two-cancer screening, organizer-related factors such as the source of notification about screening, access to screening, feedback about screening results, and promoting referrals are essential for the delivery of cancer screening [20,24,25]. Yet the corresponding data are scarce. Expanding further evidence about the effects of higher-level factors, especially at the organizer level, on two-cancer screening is critical to the design and implementation of optimal screening strategies, as well as the intervention and planned action screening programs, such as the precede-proceed model [26,27].

The aims of this study were to evaluate the attendance and determinants of organized two-cancer screening in a community-based survey conducted in rural China and to examine receivers’ perception of screening services, especially for the provider- and organizer-related factors. Our study extends this line of inquiry by examining screened women’s perspectives among the eligible women and by focusing on higher-level factors at provider and organizer levels that may only be ascertained by speaking to screened women.

## 2. Materials and Methods

### 2.1. Study Design and Participants

This was a cross-sectional study, executed door to door and face to face from June 2018 to November 2019, among participants from rural Hubei Province in central China. Among the 72 counties of Hubei Province, 12 counties were selected using purposive sampling (See Figure 1 for details), and convenience sampling was used to select 1–2 villages from each of the selected counties. All households in the selected villages were included. Inclusion and exclusion criteria in our sample are as follows: we included women aged 35 years and above that were population residents for the past six months; we excluded samples who have a hearing impairment, speech impairment, mental illness, and other serious illnesses. There were 2156 eligible respondents, of whom 41 refused the survey (1.90%). We also excluded 166 (7.70%) responses with missing key variables. Finally, a total of 1949 (90.40%) women were enrolled in this study. We obtained ethics approval from the review board of the authors’ institute [No. IORG0003571; 2019-S006]. Survey participation was voluntary, and informed consent was obtained from all participants before any data were collected. Gift incentives of about three dollars were given to each participant.

### 2.2. Measures

We operationalized our primary outcome as the attendance and willingness to participate in two-cancer screening for all eligible women and as the willingness to participate again and to recommend screening to others for screened women in organized two-cancer screening [6,9,10,22,28]. These primary outcomes were assessed using four survey items: (1) Have you participated in two-cancer screening since 2017? (response options: yes or no) (2) How willing are you to participate in two-cancer screening? (3) How willing are you to participate in two-cancer screening again? (4) How willing are you to recommend screening to others? (response options for the three previous items: very low, low, neutral, high, very high; we defined high participation willingness as responses high or very high).

The explanatory variables included aspects of the receiver, provider, and organizer of two-cancer screening. Organizer-level factors were measured with the following six items: notified about two-cancer screening (response options: yes or no, and yes had the following options: notification by GPs, friends/relatives, community women’s leaders, other community leaders); travel time to the hospital (response options: <15 min, 15–29 min, ≥30 min); received report within 2 weeks of screening; received report as of now; continuity of screening; overall perceived smoothness and usefulness of this screening program (response options of the above: yes or no) [12,20,24,29,30]. Provider-level factors were measured using six items: waiting time in the hospital (response options: <15 min, 15–29 min, ≥30 min); doctor’s explanations unclear; ward cleanliness; ward quietness; presence of others during examination; received health education (response options for the above: yes or no) [9,20,23,28]. Receiver-level factors included sociodemographic characteristics and disease-related characteristics [6,9,15,21]. The sociodemographic characteristics were: age, education status, marital status, housing characteristics as indicators of socioeconomic status, and time/day online with a mobile phone. Disease-related characteristics were: seeing doctor for gynecological problems; going to the hospital for gynecological physical examination on one’s own initiative; experiencing symptoms/discomfort of irregular vaginal bleeding, nipple discharge, breast lump, abnormal leukorrhea, or similar within the preceding 3 years; heard of two-cancer or having relatives or friends suffered from two-cancer in the past 5 years.

### 2.3. Statistical Analysis

We first analyzed the sociodemographic characteristics of all eligible women and determinants of their participation rate and participation willingness. We then conducted an in-depth analysis of screened women in organized two-cancer screening as well as the determinants of their willingness to participate again and to recommend screening to others. Finally, we analyzed the distribution characteristics of screening notifiers and their effects on willingness to participate in screening again and recommend screening to others among screened women.

We compared explanatory variables across attendance and participation willingness using χ^2^ tests. We conducted binary logistic regression analysis and calculated odds ratios (ORs) and 95% confidence intervals (CIs) in the unadjusted and adjusted regression models. All analyses were performed using IBM SPSS Statistics, version 22 (IBM Corp., Armonk, NY, USA). Statistical tests were two-tailed, and differences were considered significant with *p* < 0.05.

Sensitivity analyses were conducted to test the robustness of the results, as follows: (1) further adjustment for the effects of family support on attendance and participation willingness among all eligible women, and the willingness to participate in screening again and recommend screening to others by screened women, as well as the effects of screening notifiers on women’s willingness to participate again and to recommend screening to others; (2) using self-reported economic status instead of housing characteristics as the indicator of individual socioeconomic status for all multivariable analyses above.

### 2.4. Patient and Public Involvement

Patients or the public were not involved in the design, conduct, reporting, or dissemination plans of our research.

## 3. Results

Table 1 presents the attendance and willingness to participate in the screening of participants. Among the included women, 1278 (65.57%) did receive a screening notification. The attendance for two-cancer screening among the total participants and participants who received screening notification was 40.64% (95% CI: 38.45–42.82) and 61.19% (95% CI: 58.51–63.86), respectively; and participation willingness was 71.27% (95% CI: 69.26–73.28) and 77.15% (95% CI:74.85–79.46), respectively. Among total participants, 61.16% of participants had a primary education level or below, and 54.59% had experienced gynecological symptoms/discomforts, but only 36.33% of women took the initiative to undergo a gynecological physical examination, 14.52% reported their relatives or friends suffered from two-cancer in the last 5 years, and 71.58% had heard of two cancers. Among women who received a screening notification, the differences in the distribution characteristics were the initiative to undergo gynecological physical examinations (42.10%) and heard of two cancers (80.05%).

Table 2 presents the determinants of screening attendance and willingness among the total participants and participants who received a screening notification. In particular, after adjusting for potential confounders, participants who received a screening notification were more likely to have higher attendance and greater willingness to participate than those who did not receive a screening notification (aOR = 1.59, 95% CI: 1.27–1.99; aOR = 98.03, 95% CI: 51.44–186.82, respectively). Additionally, women who saw a doctor for gynecological problems and took the initiative to undergo a gynecological physical examination were more likely to have greater attendance (aOR = 2.15, 95% CI: 1.62–2.85; aOR = 2.15, 95% CI: 1.65–2.80, respectively) and greater screening willingness (aOR = 1.73, 95% CI: 1.31–2.29; aOR = 1.61, 95% CI: 1.24–2.09, respectively) than those without these disease-related characteristics.

Table 3 presents the results for screened women’s perceived performance of providers and organizers on organized two-cancer screening services, as well as sociodemographic and disease-related characteristics according to women’s willingness to participate again and to recommend screening to others. The proportions of the willingness to participate again and to recommend screening to others were 91.60% (95% CI: 89.61–93.59) and 86.93% (95% CI: 84.52–89.35), respectively. Satisfaction with screening services in specific domains varied substantially: 82.40% of women reported high cleanliness of the ward, whereas only 21.73% reported waiting time in the hospital < 15 min, and 82.27% of respondents reported overall perceived usefulness.

Table 4 presents determinants of the willingness to participate again and to recommend screening to others among screened women in organized two-cancer screening. In particular, after adjustment for potential confounders, all six provider-level factors showed no significant association with outcome variables. Of the five organizer-level factors, only overall perceived usefulness was significantly associated with the willingness to participate again and to recommend screening to others (aOR = 7.80, 95% CI: 4.08–14.92; aOR = 3.84, 95% CI: 2.19–6.73, respectively).

Given the effects of screening notification on the attendance and participation willingness among the total participants (Table 2), we further analyzed the effects of screening notifiers on the willingness to participate again and to recommend screening to others among screened women (Table 5). Women who were notified by GPs accounted for 8.93%, and community women’s leaders and other community leaders accounted for 46.27% and 40.00%, respectively. Women notified about screening by GPs reported 83.58% willingness to participate again; and village cadres and friends or relatives reported greater willingness to participate again, especially those notified by community women’s leaders and other community leaders (93.08% and 92.33%, respectively). A similar distribution was observed for the willingness to recommend screening to others (Table 5). After adjustment for potential confounders, compared with notification by GPs, notification by community women’s leaders and other community leaders were more likely to lead to greater willingness to participate again (aOR = 2.86, 95% CI: 1.13–7.24; aOR = 3.27, 95% CI: 1.26–8.48, respectively) and to recommend screening to others (aOR = 2.18, 95% CI: 1.02–4.65; aOR = 4.14, 95% CI: 1.84–9.30, respectively).

The results of sensitivity analyses showed no significant changes in the size and significance of the effects. Specifically, further adjustment for the effects of family support in all models of attendance and participation willingness produced results very similar to the original results (Appendix A, respectively). When we used self-reported economic status instead of housing characteristics as the indicator of individual socioeconomic status, the results were the same; that is, a positive association of screening notifiers and overall usefulness with a willingness to participate in screening was observed (Appendix A, respectively).

## 4. Discussion

To our knowledge, this was the first empirical study on the attendance of two-cancer screening using a population-based survey in rural China. The attendance and willingness to participate in screening among women who received screening notification were significantly higher than among eligible women who were not notified. Among screened women in organized two-cancer screening, being notified about screening by community leaders significantly improved women’s willingness to participate in screening and to recommend screening to others, in comparison with being notified about screening by GPs. Additionally, overall usefulness was positively associated with the willingness to participate again and to recommend screening to others.

We found moderately high performance for organizer-level factors of screening, except for continuity of screening. The proportion of women who received screening notification was less than 70%, which could be owing to China’s current social transition and population mobility; this finding also revealed the complexity and difficulty of two-cancer screening in China [12,16]. Among all eligible women, receiving notification had a significant effect on their attendance and participation willingness. This may be because, as the first step in organized two-cancer screening, notification serves as the cue or trigger to action two-cancer screening [31,32]. Furthermore, the proportion of community leaders as screening notifiers was significantly higher than that of those of GPs, which would reflect the nature of the public service of organized two-cancer screening, and the functions and importance of community leaders in providing public services and managing public affairs. Interestingly, among screened women, being notified about screening by community leaders was more conducive to improving their willingness to participate again and to recommend screening to others, in comparison with being notified by GPs. Recently, the role of lay health workers in healthcare has been emphasized, especially in public health, which may explain the effects of community leaders as notifiers on two-cancer screening [33,34]. On the one hand, women feel more respected when notified by community leaders [12]; on the other hand, as providers of public services and managers of public affairs, community leaders may be more trusted than GPs as notifiers of two-cancer screening. Recent studies show that lack of trust and inadequate performance ratings among public health agencies are a global phenomenon [35,36,37].

Our research showed that women who participated in organized two-cancer screening reported a high level of overall perceived usefulness of the screening program, and its association with the willingness to participate in screening again and to recommend screening to others was statistically significant. As the final step in organized two-cancer screening, overall perceived usefulness can help participating women gain peace of mind by dispelling doubts, anxiety, and worry about two-cancer, which would be a reinforcer for screening participation [38,39].

We also found the moderately high performance of provider-level factors in screening, except for the presence of others during screening, which accounted for more than 50% of responses. This is likely because women who go to screening together are from the same village and know each other; many of them are good friends, and their need for confidentiality might be relatively low. Interestingly, nearly all provider-level factors in screening showed no significant association with the willingness to participate again and to recommend screening to others; this could be because two-cancer screening may be perceived as a potential need and not as a medical service [12,16]. Considering free tuberculosis screening in the 1950s and nucleic acid testing for COVID-2019 during the pandemic beginning in 2020, the performance of providers in these types of services would be much better than those of other health services, especially when programs are organized and free; however, the effects on attendance by the target population are still weak [40,41]. Public health services are related to public safety, which may be the fundamental difference from medical services. Furthermore, as a part of public health services, the design and implementation of optimal cancer screening strategies (not limited to two-cancer screening) may require the involvement of the public sector (such as community leaders at the organizer level), instead of a one-man show by the health sector.

For the receiver-level factors, our study found that secondary education and having heard of two-cancer are contributing factors to screening willingness, and women with active gynecological medical examination experience will promote screening willingness and attendance. This is likely because a high level of education will have a better understanding of health knowledge and better health literacy [42]. Women who actively participate in gynecological physical examinations have higher health literacy, and they are more likely to receive health education and screening recommendations from doctors.

## 5. Limitation

First, our study population was from only one province in central China, which may lead to selection bias. Second, although this survey was conducted door to door and face to face, the sample size may be insufficient, especially in terms of opportunistic screening. Third, data were self-reported and have a probability of recall bias. Finally, our study relied on cross-sectional data and, therefore, causality cannot be established.

## 6. Conclusions

Our findings provide evidence that attendance at two-cancer screening is relatively low in rural China; however, organized two-cancer screening would be effective in improving attendance. As the first step in organized two-cancer screening, receiving screening notification would be the trigger to action for two-cancer screening, and the overall perceived usefulness of the screening program would be a reinforcer for screening participation again. Notification of women about screening by community leaders is an important organizer-level factor. As a part of public health services, the design and implementation of optimal cancer screening strategies may require public-sector involvement at the organizer level.

## Figures and Tables

**Figure 1 ijerph-19-08237-f001:**
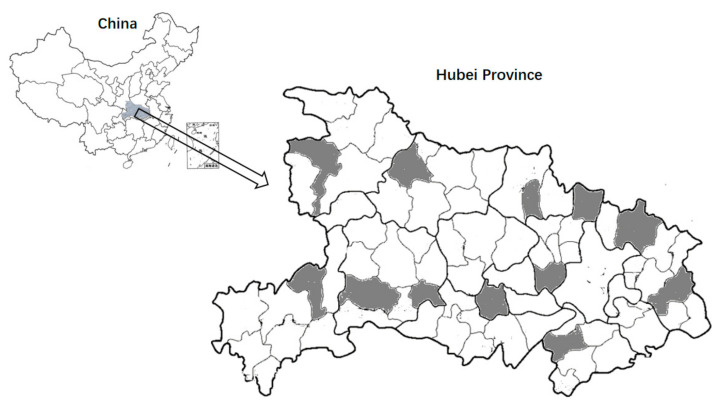
Distribution map of the sample counties.

**Table 1 ijerph-19-08237-t001:** General Characteristics of the Participants by Screening Willingness and Attendance of Cervical and Breast Cancer Screening 2018–2019.

	Total Participants									Participants Received Screening Notification								
	Total	Screening Willingness				Attendance				Total	Screening Willingness				Attendance			
	% (*n* = 1949)	Yes	No	*p*		Yes	No	*p*		% (*n* = 1278)	Yes	No	*p*		Yes	No	*p*	
**Total**	100	71.3	28.7			40.6	59.4			100	77.2	22.8			61.2	38.8		
**Organizer-level factors**																		
**Receiving screening notification**																		
Yes	65.6	71.0	52.1	<0.001		98.74	42.87	<0.001		-	-	-	-	-	-	-	-	-
No	34.4	29.0	47.9			1.26	57.13			-	-	-	-	-	-	-	-	-
**Receiver-level factors—Sociodemographic characteristics**																		
**Age**																		
65 years and above	19.0	14.7	29.8			8.8	26.0			12.7	10.8	19.2			9.0	18.6		
55–64 years	32.5	31.6	34.6			33.8	31.6			33.4	32.0	38.4			33.8	32.9		
5–54 years	33.3	36.5	25.2			40.2	28.5			37.9	39.9	31.2			40.0	34.5		
35–44 years	15.2	17.2	10.4	<0.001		17.2	13.9	<0.001		16.0	17.4	11.3	<0.001		17.3	14.1	<0.001	
**Education status**																		
Senior high school and above	8.2	8.4	7.7			8.2	8.1			8.4	8.6	7.5			7.9	9.1		
Junior high school	30.7	34.1	22.1			36.0	27.1			33.0	35.5	24.7			36.2	28.0		
Primary and lower	61.2	57.5	70.2	<0.001		55.8	64.8	<0.001		58.6	55.9	67.8	0.001		55.9	62.9	0.01	
**Marital status**																		
Married	88.3	90.5	82.7			93.6	84.6			91.4	92.9	86.3			93.6	87.9		
Single/Divorced/Widowed/Other	11.8	9.5	17.3	<0.001		6.4	15.4	<0.001		8.6	7.1	13.7	<0.001		6.4	12.1	<0.001	
**Housing characteristic**																		
Three-story house or larger	22.7	22.7	22.7			21.5	23.5			21.1	10.6	22.6			21.0	21.2		
Two-story	55.6	54.5	58.2			54.4	56.4			55.7	55.4	56.9			54.6	57.5		
One-story	12.5	13.1	11.1			13.9	11.6			13.5	13.8	12.3			14.1	12.5		
Tiled-roof house	9.2	9.7	8.0	0.317		10.2	8.6	0.203		9.8	10.2	8.2	0.61		10.4	8.9	0.639	
**Time/day online with mobile phone**																		
60 min and above	33.7	36.9	25.7			36.6	31.6			34.7	36.7	27.7			36.3	32.1		
0–59 min	66.3	63.1	74.3	<0.001		63.4	68.4	0.022		65.3	63.3	72.3	0.005		63.7	67.9	0.119	
**Receiver-level factors—Disease-related characteristics**																		
**Seeing doctor for gynecological problems**																		
Yes	30.2	35.5	17.0	<0.001		41.4	22.5	<0.001		33.0	37.8	16.8	<0.001		40.9	20.6	<0.001	
No	69.8	64.5	83.0			58.6	77.5			67.0	62.2	83.2			59.1	79.4		
**Gynecological physical examination on one’s own initiative**																		
Yes	36.3	42.2	21.8	<0.001		51.5	25.9	<0.001		42.1	47.7	23.3	<0.001		51.2	27.8	<0.001	
No	63.7	57.8	78.2			48.5	74.1			57.9	52.3	76.7			48.9	72.2		
**Experiencing symptoms/discomfort (irregular vaginal bleeding, nipple discharge, breast lump, abnormal leukorrhea, or similar)**																		
Yes	54.6	57.7	46.8	<0.001		53.7	55.2	0.495		55.3	57.3	48.6	0.009		54.1	57.3	0.267	
No	45.4	42.3	53.2			46.3	44.8			44.7	42.7	51.4			45.9	42.7		
**Heard of two-cancer**																		
Yes	71.6	76.8	58.6	<0.001		82.6	64.0	<0.001		80.1	83.2	69.5	<0.001		82.5	76.2	0.006	
No	28.4	23.2	41.4			17.4	36.0			20.0	16.8	30.5			17.5	23.8		
**Relatives or friends suffered from two-cancer in the past 5 years**																		
Yes	14.5	15.6	12.0	0.042		15.0	14.2	0.601		15.0	15.6	13.0	0.274		14.8	15.3	0.812	
No	85.5	84.5	88.0			85.0	85.8			85.0	84.4	87.0			85.2	84.7		

**Table 2 ijerph-19-08237-t002:** Determinants of screening willingness and attendance of breast and cervical cancer screening in rural China 2018–2019.

	Total Participants				Participants Received Screening Notification			
	Screening Willingness		Attendance		Screening Willingness		Attendance	
	Unadjusted OR (95%CI)	Adjusted OR (95%CI)	Unadjusted OR (95%CI)	Adjusted OR (95%CI)	Unadjusted OR (95%CI)	Adjusted OR (95%CI)	Unadjusted OR (95%CI)	Adjusted OR (95%CI)
**Organizer-level factors**								
**Receiving screening notification**								
Yes	2.25 (1.83,2.75) ***	1.59 (1.27,1.99) ***	104.21 (55.25,196.56) ***	98.03 (51.44,186.82) ***				
No	1.00	1.00	1.00	1.00				
**Receiver-level factors—Sociodemographic characteristics**								
**Age**								
65 years and above	0.30 (0.21,0.42) ***	0.53 (0.35,0.81) **	0.28 (0.20,0.39) ***	0.60 (0.37,0.99) *	0.36 (0.22,0.60) ***	0.59 (0.33,1.06)	0.40 (0.26,0.60) ***	0.58 (0.35,0.97) *
55–64 years	0.55 (0.39,0.77) ***	0.78 (0.54,1.12)	0.87 (0.66,1.15)	1.25 (0.85,1.85)	0.54 (0.35,0.83) **	0.79 (0.49,1.27)	0.84 (0.59,1.19)	1.17 (0.78,1.74)
45–54 years	0.87 (0.62,1.23)	0.91 (0.64,1.31)	1.14 (0.87,1.50)	1.13 (0.79,1.63)	0.83 (0.54,1.28)	0.92 (0.58,1.46)	0.95 (0.67,1.34)	1.06 (0.73,1.54)
35–44 years	1.00	1.00	1.00	1.00	1.00	1.00	1.00	1.00
**Education status**								
Senior high school and above	1.33 (0.92,1.92)	0.82 (0.55,1.24)	1.17 (0.84,1.64)	0.75 (0.48,1.18)	1.39 (0.85,2.28)	0.95 (0.55,1.63)	0.98 (0.65,1.48)	0.71 (0.45,1.11)
Junior high school	1.88 (1.49,2.37) ***	1.39 (1.07,1.81) *	1.55 (1.27,1.89) ***	1.19 (0.91,1.57)	1.75 (1.29,2.36) ***	1.45 (1.04,2.03) *	1.45 (1.13,1.87) **	1.24 (0.94,1.64)
Primary and lower	1.00	1.00	1.00	1.00	1.00	1.00	1.00	1.00
**Marital status**								
Married	2.00 (1.50,2.65) ***	1.20 (0.87,1.65)	2.64 (1.91,3.66) ***	1.33 (0.87,2.04)	2.08 (1.38,3.14) ***	1.36 (0.86,2.15)	2.02 (1.36,2.99) ***	1.36 (0.88,2.09)
Single/Divorced/Widowed/Other	1.00	1.00	1.00	1.00	1.00	1.00	1.00	1.00
**Housing characteristic**								
Three-story house or larger	0.83 (0.56,1.23)	0.59 (0.38,0.91) *	0.76 (0.54,1.09)	0.74 (0.46,1.20)	0.73 (0.43,1.24)	0.55 (0.32,0.97) *	0.85 (0.55,1.32)	0.69 (0.43,1.12)
Two-story	0.77 (0.54,1.11)	0.60 (0.40,0.88) **	0.81 (0.59,1.11)	0.71 (0.46,1.08)	0.78 (0.49,1.26)	0.64 (0.39,1.07)	0.81 (0.55,1.21)	0.70 (0.45,1.07)
One-story	0.98 (0.63,1.53)	0.82 (0.51,1.32)	1.00 (0.68,1.48)	0.92 (0.55,1.53)	0.90 (0.50,1.60)	0.82 (0.45,1.50)	0.96 (0.60,1.56)	0.93 (0.55,1.55)
Tiled-roof house	1.00	1.00	1.00	1.00	1.00	1.00	1.00	1.00
**Time/day online with a mobile phone**								
60 min and above	1.69 (1.36,2.10) ***	1.21 (0.94,1.56)	1.25 (1.03,1.51) *	1.00 (0.76,1.31)	1.51 (1.13,2.01) **	1.06 (0.76,1.48)	1.21 (0.95,1.53)	0.97 (0.73,1.28)
0–59 min	1.00	1.00	1.00	1.00	1.00	1.00	1.00	1.00
**Receiver-level factors—Disease-related characteristics**								
**Seeing doctor for gynecological problems**								
Yes	2.69 (2.11,3.45) ***	1.73 (1.31,2.29) ***	2.44 (2.00,2.97) ***	2.15 (1.62,2.85) ***	3.02 (2.16,4.21) ***	2.02 (1.41,2.91) ***	2.68 (2.06,3.47) ***	2.04 (1.53,2.72) ***
No	1.00	1.00	1.00	1.00	1.00	1.00	1.00	1.00
**Gynecological physical examination on one’s own initiative**								
Yes	2.620 (2.09,3.29) ***	1.61 (1.24,2.09) ***	3.04 (2.51,3.68) ***	2.15 (1.65,2.80) ***	3.00 (2.23,4.05) ***	2.09 (1.51,2.90) ***	2.72 (2.13,3.46) ***	2.10 (1.61,2.75) ***
No	1.00	1.00	1.00	1.00	1.00	1.00	1.00	1.00
**Experiencing symptoms/discomfort (irregular vaginal bleeding, nipple discharge, breast lump, abnormal leukorrhea, or similar)**								
Yes	1.55 (1.28,1.89) ***	1.45 (1.17,1.79) ***	0.94 (0.78,1.13)	0.75 (0.59,0.96) *	1.42 (1.09,1.84) **	1.33 (1.01,1.75) *	0.88 (0.70,1.10)	0.80 (0.63,1.02)
No	1.00	1.00	1.00	1.00	1.00	1.00	1.00	1.00
**Heard of two-cancer**								
Yes	2.34 (1.90,2.89) ***	1.59 (1.26,2.01) ***	2.66 (2.14,3.31) ***	1.32 (0.98,1.78)	2.17 (1.61,2.92) ***	1.69 (1.22,2.34) **	1.47 (1.11,1.94) **	1.26 (0.93,1.71)
No	1.00	1.00	1.00	1.00	1.00	1.00	1.00	1.00
**Relatives or friends suffered from two-cancer in the past 5 years**								
Yes	1.36 (1.01,1.82) *	1.00 (0.73,1.37)	1.07 (0.83,1.38)	0.83 (0.60,1.16)	1.24 (0.85,1.81)	0.92 (0.61,1.39)	0.96 (0.70,1.32)	0.82 (0.58,1.14)
No	1.00	1.00	1.00	1.00	1.00	1.00	1.00	1.00

Note: * *p* < 0.05, ** *p* < 0.01, *** *p* < 0.001.

**Table 3 ijerph-19-08237-t003:** The willingness to participate again and to recommend screening to others of organized breast and cervical cancer screening in rural China 2018–2019.

	Total	Willingness to Participate Again				Willingness to Recommend Screening to Others			
	% (*n* = 750)	Yes	No	*p*		Yes	No	*p*	
**Total**	100	91.6	8.4			86.9	13.1		
**Organizer-lever factors**									
**Travel time to the hospital**									
≥30 min	30.5	30.3	33.3	0.38		30.4	31.6	0.209	
15–29 min	17.5	18.1	11.1			18.4	11.2		
<15 min	52.0	51.7	55.6			51.2	57.1		
**Received report within 2 weeks of screening**									
Yes	60.5	60.4	61.9	0.816		60.4	61.2	0.881	
No	39.5	39.6	38.1			39.6	38.8		
**Continuity of screening**									
Yes	37.3	37.7	33.3	0.493		39.0	26.5	0.018	
No	62.7	62.3	66.7			61.0	73.5		
**Received report as of now**									
Yes	83.5	83.6	82.5	0.836		83.1	85.7	0.521	
No	16.5	16.5	17.5			16.9	14.3		
**Overall perceived smoothness**									
Yes	78.0	79.6	60.3	<0.001		79.3	69.4	0.027	
No	22.0	20.4	39.7			20.7	30.6		
**Overall perceived usefulness**									
Yes	82.3	85.7	44.4	<0.001		85.3	62.2	<0.001	
No	17.7	14.3	55.6			14.7	37.8		
**Provider-lever factors**									
**Waiting time in the hospital**									
≥30 min	69.5	69.6	68.3	0.969		70.3	64.3	0.49	
15–29 min	8.8	8.7	9.5			8.6	10.2		
<15 min	21.7	21.7	22.2			21.2	25.5		
**Doctor’s explanations unclear**									
Very low/Somewhat low	68.8	69.0	66.7	0.091		68.6	70.4	0.141	
Neutral	6.4	5.8	12.7			5.8	10.2		
Somewhat high/Very high	24.8	25.2	20.6			25.6	19.4		
**Ward cleanliness**									
Somewhat high/Very high	82.4	82.4	82.5	0.175		82.2	83.7	0.909	
Neutral	15.7	16.0	12.7			16.0	14.3		
Very low/Somewhat low	1.9	1.6	4.8			1.8	2.0		
**Ward quietness**									
Somewhat high/Very high	74.0	73.8	76.2	0.181		72.9	81.6	0.073	
Neutral	14.7	15.3	7.9			15.8	7.1		
Very low/Somewhat low	11.3	10.9	15.9			11.4	11.2		
**Presence of others during examination**									
No	43.9	44.0	42.9	0.866		43.9	43.9	0.998	
Yes	56.1	56.0	57.1			56.1	56.1		
Received health education									
Yes	60.5	61.1	54.0	0.265		62.7	45.9	0.001	
No	39.5	38.9	46.0			37.3	54.1		
**Receiver-lever factors—Sociodemographic characteristics**									
**Age**									
65 years and above	8.5	8.4	9.5			8.3	10.2		
55–64 years	33.7	33.2	39.7			31.6	48.0		
45–54 years	40.1	40.3	38.1			41.3	32.7		
35–44 years	17.6	18.1	12.7	0.613		18.9	9.2	0.004	
**Education status**									
Senior high school and above	8.0	8.3	4.8			8.4	5.1		
Junior high school	36.0	37.0	25.4			36.8	30.6		
Primary and lower	56.0	54.7	69.8	0.067		54.8	64.3	0.176	
**Marital status**									
Married	93.3	93.5	92.1			94.0	88.8		
Single/Divorced/Widowed/Other	6.7	6.6	7.9	0.673		6.0	11.2	0.052	
**Housing characteristic**									
Three-story house or larger	21.5	21.0	27.0			21.4	21.5		
Two-story	54.0	53.7	57.1			54.5	51.0		
One-story	14.1	14.6	9.5			14.0	15.3		
Tiled-roof house	10.4	10.8	6.4	0.359		10.1	12.2	0.882	
**Time/day online with a mobile phone**									
60 min and above	36.7	37.0	33.3			37.9	28.6		
0–59 min	63.3	63.0	66.7	0.566		62.1	71.4	0.074	
**Receiver-lever factors—Disease-related characteristics**									
**Seeing doctor for gynecological problems**									
Yes	40.9	41.8	31.8	0.121		41.6	36.7	0.365	
No	59.1	58.2	68.3			58.4	63.3		
**Gynecological physical examination on one’s own initiative**									
Yes	51.1	51.5	46.0	0.404		52.2	43.9	0.127	
No	48.9	48.5	54.0			47.9	56.1		
**Experiencing symptoms/discomfort (irregular vaginal bleeding, nipple discharge, breast lump, abnormal leukorrhea, or similar)**									
Yes	53.6	53.4	55.6	0.745		54.3	49.0	0.325	
No	46.4	46.6	44.4			45.7	51.0		
**Heard of two-cancer**									
Yes	82.3	83.7	66.7	0.001		85.3	62.2	<0.001	
No	17.7	16.3	33.3			14.7	37.8		
**Relatives or friends suffered from two-cancer in the past 5 years**									
Yes	14.8	15.6	6.4	0.048		16.1	6.1	0.009	
No	85.2	84.4	93.7			83.9	93.9		
**History of previous screening**									
Yes	61.2	62.9	42.9	0.002		63.2	48.0	0.004	
No	38.8	37.1	57.1			36.8	52.0		

**Table 4 ijerph-19-08237-t004:** Determinants of the willingness to participate again and to recommend screening to others of organized breast and cervical cancer screening in rural China 2018–2019.

	Willingness to Participate Again		Willingness to Recommend Screening to Others	
	Unadjusted OR (95%CI)	Adjusted OR (95%CI)	Unadjusted OR (95%CI)	Adjusted OR (95%CI)
**Organizer-lever factors**				
**Travel time to the hospital**				
≥30 min	0.98 (0.55, 1.72)	0.81 (0.40, 1.65)	1.07 (0.67, 1.72)	1.05 (0.59, 1.87)
15–29 min	1.75 (0.76, 4.03)	1.46 (0.56, 3.78)	1.83 (0.93, 3.61)	1.70 (0.79, 3.64)
<15 min	1.00	1.00	1.00	1.00
**Received report within 2 weeks of screening**				
Yes	0.94 (0.55, 1.60)	0.77 (0.39, 1.53)	0.97 (0.63, 1.50)	0.91 (0.53, 1.56)
No	1.00	1.00	1.00	1.00
**Received report as of now**				
Yes	1.08 (0.54, 2.12)	0.93 (0.38, 2.26)	0.82 (0.45, 1.50)	0.60 (0.28, 1.26)
No	1.00	1.00	1.00	1.00
**Continuity of screening**				
Yes	1.21 (0.70, 2.09)	0.88 (0.44, 1.74)	1.77 (1.10, 2.84) *	1.44 (0.82, 2.52)
No	1.00	1.00	1.00	1.00
**Overall perceived smoothness**				
Yes	2.57 (1.50, 4.40) ***	1.57 (0.80, 3.10)	1.69 (1.06, 2.70) *	1.35 (0.76, 2.38)
No	1.00	1.00	1.00	1.00
**Overall perceived usefulness**				
Yes	7.51 (4.37, 12.91) ***	7.80 (4.08, 14.92) ***	3.51 (2.13, 5.58) ***	3.84 (2.19, 6.73) ***
No	1.00	1.00	1.00	1.00
**Provider-lever factors**				
**Waiting time in the hospital**				
≥30 min	1.04 (0.56, 1.96)	1.11 (0.52, 2.37)	1.32 (0.80, 2.17)	1.30 (0.72, 2.33)
15–29 min	0.94 (0.35, 2.56)	1.01 (0.32, 3.18)	1.01 (0.46, 2.25)	0.85 (0.35, 2.06)
<15 min	1.00	1.00	1.00	1.00
**Doctor’s explanations unclear**				
Very low/Somewhat low	0.85 (0.45, 1.62)	1.00 (0.47, 2.10)	0.74 (0.43, 1.26)	0.64 (0.34, 1.18)
Neutral	0.38 (0.15, 0.97) *	0.72 (0.23, 2.31)	0.43 (0.19, 1.00)	0.62 (0.23, 1.67)
Somewhat high/Very high	1.00	1.00	1.00	1.00
**Ward cleanliness**				
Somewhat high/Very high	3.75 (0.87, 16.22)	5.58 (0.92, 33.94)	1.24 (0.25, 6.12)	0.97 (0.16, 5.93)
Neutral	2.97 (0.80, 10.98)	3.50 (0.64, 19.13)	1.09 (0.24, 4.96)	0.98 (0.17, 5.65)
Very low/Somewhat low	1.00	1.00	1.00	1.00
**Ward quietness**				
Somewhat high/Very high	2.80 (0.92, 8.53)	2.20 (0.59, 8.17)	2.19 (0.81, 5.91)	2.28 (0.73, 7.07)
Neutral	1.41 (0.68, 2.90)	1.21 (0.47, 3.09)	0.88 (0.45, 1.74)	0.77 (0.34, 1.74)
Very low/Somewhat low	1.00	1.00	1.00	1.00
**Presence of others during examination**				
No	1.05 (0.62, 1.76)	1.11 (0.61, 2.02)	1.00 (0.65, 1.53)	1.19 (0.73, 1.93)
Yes	1.00	1.00	1.00	1.00
**Received health education**				
Yes	1.34 (0.80, 2.25)	0.95 (0.49, 1.85)	1.98 (1.29, 3.04) **	1.50 (0.89, 2.54)
No	1.00	1.00	1.00	1.00
**Receiver-lever factors—Sociodemographic characteristics**				
**Age**				
65 years and above	0.62 (0.21, 1.88)	1.04 (0.25, 4.41)	0.40 (0.15, 1.03)	0.71 (0.22, 2.26)
55–64 years	0.59 (0.26, 1.34)	0.62 (0.22, 1.73)	0.32 (0.15, 0.68) **	0.38 (0.16, 0.92) *
45–54 years	0.75 (0.33, 1.70)	0.93 (0.36, 2.41)	0.62 (0.29, 1.33)	0.68 (0.30, 1.58)
35–44 years	1.00	1.00	1.00	1.00
**Education status**				
Senior high school and above	2.22 (0.67, 7.40)	2.39 (0.59, 9.65)	1.94 (0.75, 5.04)	1.31 (0.43, 3.99)
Junior high school	1.86 (1.03, 3.36) *	1.57 (0.76, 3.23)	1.41 (0.89, 2.25)	0.93 (0.53, 1.62)
Primary and lower	1.00	1.00	1.00	1.00
**Marital status**				
Married	1.23 (0.47, 3.22)	1.07 (0.34, 3.31)	1.99 (0.98, 4.03)	1.62 (0.70, 3.71)
Single/Divorced/Widowed/Other	1.00	1.00	1.00	1.00
**Housing characteristic**				
Three-story house or larger	0.46 (0.15, 1.41)	0.38 (0.11, 1.34)	1.21 (0.56, 2.61)	1.45 (0.60, 3.52)
Two-story	0.55 (0.19, 1.60)	0.53 (0.16, 1.76)	1.29 (0.65, 2.56)	1.58 (0.72, 3.47)
One-story	0.90 (0.25, 3.31)	0.84 (0.20, 3.49)	1.10 (0.49, 2.51)	1.06 (0.42, 2.67)
Tiled-roof house	1.00	1.00	1.00	1.00
**Time/day online with a mobile phone**				
60 min and above	1.17 (0.68, 2.03)	1.01 (0.51, 1.99)	1.53 (0.96, 2.43)	1.16 (0.66, 2.04)
0–59 min	1.00	1.00	1.00	1.00
**Receiver-lever factors—Disease-related characteristics**				
**Seeing a doctor for gynecological problems**				
Yes	1.54 (0.89, 2.68)	1.89 (0.96, 3.73)	1.23 (0.79, 1.90)	0.96 (0.56, 1.64)
No	1.00	1.00	1.00	1.00
**Gynecological physical examination on one’s own initiative**				
Yes	1.25 (0.74, 2.09)	0.94 (0.50, 1.76)	1.39 (0.91, 2.14)	1.18 (0.70, 1.97)
No	1.00	1.00	1.00	1.00
Yes	0.92 (0.55, 1.54)	0.88 (0.48, 1.61)	1.24 (0.81, 1.89)	1.07 (0.66, 1.74)
No	1.00	1.00	1.00	1.00
**Heard of two-cancer**				
Yes	2.57 (1.46, 4.50) ***	1.69 (0.86, 3.32)	3.51 (2.21, 5.58) ***	2.30 (1.34, 3.94) **
No	1.00	1.00	1.00	1.00
**Relatives or friends suffered from two-cancer in the past 5 years**				
Yes	2.72 (0.97, 7.65)	2.20 (0.71, 6.78)	2.94 (1.26, 6.90) *	2.23 (0.90, 5.55)
No	1.00	1.00	1.00	1.00
**History of previous screening**				
Yes	2.26 (1.34, 3.81) **	1.87 (0.98, 3.57)	1.86 (1.22, 2.86) **	1.50 (0.90, 2.50)
No	1.00	1.00	1.00	1.00

Note: * *p* < 0.05, ** *p* < 0.01, *** *p* < 0.001.

**Table 5 ijerph-19-08237-t005:** The association of the resource of notification about screening with the willingness to participate again and to recommend screening to others of organized breast and cervical cancer screening in rural China 2018–2019.

Resource of Notification About Screening	*n* (%)	Willingness to Participate Again		Willingness Recommend Screening to Others	
		%	Adjusted OR (95%CI)	%	Adjusted OR (95%CI)
Other community leaders	300 (40.00)	92.33	3.27 (1.26, 8.48) *	90.00	4.14 (1.84, 9.30) ***
Community women’s leaders	347 (46.27)	93.08	2.86 (1.13, 7.24) *	86.17	2.18 (1.02, 4,65) *
Friends or relatives	36 (4.80)	85.29	1.50 (0.37, 6.17)	88.24	3.50 (0.91, 13.47)
GPs	67 (8.93)	83.58	1.00	76.12	1.00

Note: control variables; travel time to the hospital, received report within 2 weeks of screening, received report as of now, continuity of screening, overall perceived smoothness, overall perceived usefulness, waiting time in the hospital, doctor’s explanations unclear, ward cleanliness, ward quietness, presence of others during examination, received health education, age, education status, marital status, housing characteristic, time/day online with a mobile phone, seeing a doctor for gynecological problems, gynecological physical examination on one’s own initiative, experiencing symptoms/discomfort, heard of two-cancer, relatives or friends suffered from two-cancer in the past 5 years, history of previous screening; * *p* < 0.05, *** *p* < 0.001.

## Data Availability

The study database is available via e-mail to the corresponding authors: Yuan Liang (liangyuan217@hust.edu.cn).

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
