# Peer review of "Organized Breast and Cervical Cancer Screening: Attendance and Determinants in Rural China"

_ijerph, 2022, doi:10.3390/ijerph19148237_

Round 1

Reviewer 1 Report

Thank you for your submission.

Kindly cut short your material and methods

Kindly expand the discussion part in relevance to your finding. 

Author Response

Dear reviewer:

Thank you for your quick turnaround and the critical reviews, which helped us greatly improve the quality of our paper. We have revised the article according to your suggestions, the attachment is a reply to your question, please check it out, thank you!

Reviewer 2 Report

Thank you for the opportunity to review this manuscript. The topic is very timely. Covid 19 pandemic has had a significant negative effect on oncology screenings, but cancer incidence has increased.
However, I have some suggestions to improve this research paper. 

The introduction talks about determinants of organized cancer screening. In this context, I think it’s very important to introduce the Precede-proceed model. It assumes that health and health behaviors are influenced by multiple factors - epidemiological, socio-psychological, administrative, political, environmental – which must be considered and assessed for their modifiability to ensure effective interventions and it helps planning action to improve screening programs. 
Here are some references “Saulle R, Sinopoli A, De Paula Baer A, Mannocci A, Marino M, De Belvis AG, Federici A, La Torre G. The PRECEDE-PROCEED model as a tool in Public Health screening: a systematic review. Clin Ter. 2020 Mar-Apr;171(2):e167-e177. doi: 10.7417/CT.2020.2208. PMID: 32141490”, “Cereda D, Federici A, Guarino A, Serantoni G; Gruppo PRECEDE-PROCEED, Coppola L, Lemma P, Rossi PG. Development and first application of an audit system for screening programs based on the PRECEDE-PROCEED model: an experience with breast cancer screening in the region of Lombardy (Italy). BMC Public Health. 2020 Nov 25;20(1):1778. doi: 10.1186/s12889-020-09842-8. PMID: 33238924; PMCID: PMC7687705”.

Methods are well structured.

In the discussion, I think I suggest talking about the importance of health literacy level in screening participation. Reference “Berkman ND, Sheridan SL, Donahue KE, Halpern DJ, Crotty K. Low health literacy and health outcomes: an updated systematic review. Ann Intern Med. 2011 Jul 19;155(2):97-107. doi: 10.7326/0003-4819-155-2-201107190-00005. PMID: 21768583.”

Author Response

(The authors gave the same response as above.)

Reviewer 3 Report

ijerph-1716854:  Organized breast and cervical cancer screening: Attendance and determinants in rural China.

The authors have examined a number of factors, at several levels (organizer, provider, receiver), on women’s participation in “two-cancer” (cervical and breast) screening in rural counties of Hubei Province.  The questions and results of this study are of importance in the context of the highly prescribed and particular social, medical/public health, and political hierarchies of village organization and leadership in rural China.  Given the rising rates of ovarian and breast cancers, especially among younger women, and the fairly low level of two cancer screening in rural Chinese communities, it is critical to determine factors that will both encourage women to participate in screening programs and to do so with positive engagement and understanding of the importance of early screening procedures.

METHODS:

As the authors acknowledge, much of the literature to date has focused on demographic characteristics of the “receiver”, i.e., the women who opt either to undergo screening or not.  The goals of this study are to expand the inquiry to understand community and communication factors that may also influence a woman’s decision about screening engagement.  The study used a survey of women, aged 35 or older, from selected villages within 12 of 72 counties in the Province. Of the 1949 participants, about 2/3 had received some kind of two-cancer screening notification, while the other third had not.  Of those who had received notification, a sizeable proportion (61.19%) underwent cancer screening.  For those who did not receive notification, only about 1.5% underwent screening.  These results certainly support the importance of some sort of notification in facilitating participation in two-cancer screening – the ultimate goal of this section of the national public health services program.

In elucidating non-“receiver” factors potentially affecting screening attendance, sometimes odd levels of detail were examined.  For instance, one provider level factor was waiting time at the hospital, with responses coded as “<15 min, 15-29 min, >30 min”. Given that this is a retrospective survey, that level of detail may be difficult to recall accurately.  On the other hand, one of the disease related characteristics was, “experiencing symptoms/discomfort of irregular vaginal bleeding, nipple discharge, breast lump, abnormal leukorrhea or similar within the past 3 years”. This seems to be an overly large category that might have at least been broken down into vaginal and breast concerns.

Nevertheless, it should be acknowledged that the authors sought information about a wide variety of issues, at many levels.  And their particular choices of questions asked may well reflect the particularities of knowledge, culture and concerns among rural Chinese women (of which I am not an expert).

RESULTS:

In terms of the presentation of the results, unfortunately the most salient results (the effects of notification on screening and willingness and attendance), are somewhat hidden in the text early in the results section.  And then there are a series of detailed tables which examine the results by all of the various receiver, provider and organizer variables examined in the study. The tables give loads of general information, but it is unclear that it is needed to have the data for both the ‘Total participants’ and the ‘Participants received screening notification’ In Tables I & II as, almost universally, the distributions across the variable categories were very similar in both groups.  This could well be in part due to the much higher representation of women receiving notification in the total, but it also indicates that women not receiving notification were not hugely different in their profiles.  This point could easily and clearly made in the text, simplifying these two tables by cutting them in half.

Tables 3 and 4 are duplicative of each other, with different statistical outcomes being presented.  These should be consolidated into a single table, perhaps offering the Y/N results as shown in Table 3, with asterisks indicating significance of Chi-squared test.  Actual Chi-squared values are not needed.  That would leave room for a column with the aOR and 95% confidence levels.

Similarly, Figure 2 and Table 5 have duplicated information, presented in different ways.  Given the detail of data presentation in the text, perhaps Figure 2 would suffice, with some indication of number (%) given for each bar.  Or the bar graph might be reconstructed to first have total height reflecting the number (%) of participants in each group with shading within those bars to indicate the proportion of those women willing to participate again or recommend screening to others.

These suggestions are recognition that reading this manuscript through several times, I kept being overwhelmed by the data presentation which was very explicit in the text, and then redundant in the tables and figures.  The manuscript would be greatly improved, and the main messages of the results much clearer and cleaner, if there was considerable shrinking of the data presentation.   If the authors and the journal believe all these results should be available to readers, then the use of Supplementary data (appendices) should be considered.

DISCUSSION:

The second paragraph of the discussion was fascinating, as it gave social and political context to the role of public health vs. community leader communications about screening.  I am not sure if, in other societies/cultures, medical doctors (GPs) and public health officials would be grouped together in terms of public opinion and trust.  I would remove the final sentence of that paragraph. It is made particularly confusing by the last few (fascinating, also) comments on public health as being related to public safety, not medical care., in rural China.

GENERAL EDITING:

While limited editing throughout the manuscript would improve the presentation, one clear change is the need to replace “one-man show by health sector”, a phrase that occurs many times in the paper.

Author Response

(The authors gave the same response as above.)

Reviewer 4 Report

This manuscript deals with a topic (attendance of life-saving screening programs in the most populous country in the world) whose importance cannot be overstated. However, it has some limitations that should be addressed (and possibly fixed) or acknowledged. Below a list of suggestions to improve the manuscript:

1) Introduction: in the first part, it is a bit confusing to see all those figures, %s etc. for the cervical and breast cancer programs taken together. In many countries, those programs target women in different age ranges and it would be better to give separate figures. 

2) Introduuction: in general, it should be shortened. There is nothing wrong, but it turns out too long and of difficult reading. 

3) Methods: lines 106-113 are actually Results and should be moved in that section (while avoiding duplications).

4) Results: the sentence in lines 181-183 is unclear, please rephrase. 

5) Tables: please drop X2 values (corresponding p-values are enough) and limit to the first decimal figure. The tables are too crowded now and discourage a careful examination. 

6) Table 2: The results for the Education status are a little strange (at least, unexpected to me not to see a trend), and should be commented more in the Results and in the Discussion.

7) Discussion: I'm not Chinese so cultural differences may make my understanding difficult, but for Western European countries (where I live), it is strange to hear that "community leaders may be more trusted than GPs". I'm not attaching any judgment to that (i.e. I'm not saying it is good or not, it works or it doesn't, etc) but I would like to hear more about that. 

8) Discussion, lines 306-308: is that an interpretation of the Authors? Because I can't find anything in the data supporting this statement. If it is an educated guess by the Authors, that should be made clear.

9) Limitation: "the sample may represent the average level in China"... this is a very problematic statement. The sampling is completely non-systematic (purposive + convenience, i.e. twice non-systematic), and I would say there is no way the Authors can infer about generalizability to the whole of China. So this is the most important limitations and it should be said very clearly, while the current statement (line326) is completely unjustified and should be removed. 

Author Response

(The authors gave the same response as above.)
